# Exploring the Multimodal Role of *Yucca schidigera* Extract in Protection against Chronic Ammonia Exposure Targeting: Growth, Metabolic, Stress and Inflammatory Responses in Nile Tilapia (*Oreochromis niloticus* L.)

**DOI:** 10.3390/ani11072072

**Published:** 2021-07-12

**Authors:** Zizy I. Elbialy, Abdallah S. Salah, Ahmed Elsheshtawy, Merna Rizk, Muyassar H. Abualreesh, Mohamed M. Abdel-Daim, Shimaa M. R. Salem, Ahmad El Askary, Doaa H. Assar

**Affiliations:** 1Department of Fish Processing and Biotechnology, Faculty of Aquatic and Fisheries Sciences, Kafrelsheikh University, Kafrelsheikh 33516, Egypt; ahmed_elsheshtawy@fsh.kfs.edu.eg (A.E.); mernarezk27@gmail.com (M.R.); 2Department of Aquaculture, Faculty of Aquatic and Fisheries Sciences, Kafrelsheikh University, KafrelSheikh 33516, Egypt; 3Institute of Aquaculture, University of Stirling, Stirling FK9 4LA, UK; 4Department of Marine Biology, Faculty of Marine Sciences, King Abdul-Aziz University (KAU), Jeddah 21589, Saudi Arabia; Mabulreesh1@kau.edu.sa; 5Department of Zoology, College of Science, King Saud University, P.O. Box 2455, Riyadh 11451, Saudi Arabia; abdeldaim.m@vet.suez.edu.eg; 6Pharmacology Department, Faculty of Veterinary Medicine, Suez Canal University, Ismailia 41522, Egypt; 7Department of Animal Nutrition and Nutritional Deficiency Diseases, Faculty of Veterinary Medicine, Mansoura University, Mansoura 35516, Egypt; shimaaradi_2009@mans.edu.eg; 8Department of Clinical Laboratory Sciences, College of Applied Medical Sciences, Taif University, P.O. Box 11099, Taif 21944, Saudi Arabia; a.elaskary@tu.edu.sa; 9Clinical Pathology Department, Faculty of Veterinary Medicine, Kafrelsheikh University, KafrelSheikh 33516, Egypt

**Keywords:** *Yucca schidigera*, ammonia toxicity, tilapia, antioxidant, growth, gene expression

## Abstract

**Simple Summary:**

Ammonia is a problematic environmental toxicant for aquatic species. The current study aimed to declare the modulatory effect(s) of YSE against chronic ammonia intoxication in Nile tilapia through its effects on growth performance, haemato-biochemical and antioxidant-related parameters, and histopathological changes, as well as the molecular gene expression of some genes related to appetite and growth, glucose and lipid metabolism and some inflammatory cytokines. Our results indicated that *Yucca schidigera* extract alleviated the adverse impacts induced by ammonia intoxication. YSE could be used as a functional water supplement in aquaculture.

**Abstract:**

Ammonia is a critical hazardous nitrogen metabolic product in aquaculture. Despite trials for its control, ammonia intoxication remains one of the most critical issues to overcome. In this study, we explored the modulatory effect and potential mechanism by which *Yucca schidigera* extract (YSE) can ameliorate ammonia intoxication-induced adverse effects on tilapia health and metabolism. A total number of 120 Nile tilapia were evenly assigned into four groups with three replicates each. The first group served as normal control group; the second group was exposed to ammonia alone from the beginning of the experiment and for four weeks. The third group was supplied with YSE in water at a dose of 8 mg/L and exposed to ammonia. The fourth group was supplied with YSE only in water at a dose of 8 mg/L. YSE supplementation succeeded in improving water quality by reducing pH and ammonia levels. Moreover, YSE supplementation markedly alleviated chronic ammonia-induced adverse impacts on fish growth by increasing the final body weight (FBW), specific growth rate (SGR), feed intake and protein efficiency ratio (PER) while reducing the feed conversion ratio (FCR) via improvements in food intake, elevation of hepatic insulin-like growth factor (*ILGF-1)* and suppression of myostatin (*MSTN)* expression levels with the restoration of lipid reserves and the activation of lipogenic potential in adipose tissue as demonstrated by changes in the circulating metabolite levels. In addition, the levels of hepato-renal injury biomarkers were restored, hepatic lipid peroxidation was inhibited and the levels of hepatic antioxidant biomarkers were enhanced. Therefore, the current study suggests that YSE supplementation exerted an ameliorative role against chronic ammonia-induced oxidative stress and toxic effects due to its free radical-scavenging potential, potent antioxidant activities and anti-inflammatory effects.

## 1. Introduction

Increased demand for fish as food source is a consequence of pervasive population growth [1]. Fish farming is an ideal potential means of facing the high food and nutrition demands of human beings [2]. However, the intensification of aquaculture in turn causes water pollution, which is consistently related to increased levels of ammonia, and represents about 70% of nitrogenous fish wastes [3,4]. Decomposed un-eaten feed in water favors ammonia formation, and thus subjects fish to stress risk [5]. In addition to sewage effluents, agricultural run-off and industrial wastes are major sources that are attributed to elevating ammonia levels in water bodies [6].

Ammonia is measured as total ammonia nitrogen (TAN), which is calculated as the sum of both un-ionized ammonia (NH_3_) and ionized (NH4^+^) forms [7]. NH_3_ can easily diffuse across the gill membranes due to its lipid solubility and lack of charge; therefore NH_3_ is considered 300–400 times more toxic than NH4^+^ [8]. Moreover, Lemarie et al. [9] mentioned that ammonia is stressful for fish even at low concentrations and its effect is dose dependent. High ammonia levels in water cause retardation of fish growth, tissue erosion and degeneration, immune suppression and increased mortality [10]. Ammonia induces oxidative stress via the overproduction of ROS (reactive oxygen species) [11], which deteriorates important biomolecules, such as DNA, proteins and lipids, and initiates a cascade of events that causes impairment of cellular functions [12].

Stress is a situation that threatens homeostasis in which subsequent alterations of the metabolic pathways occur to meet the host demands [13]. Chronic stress is caused by several adverse environmental conditions which suppress fish somatic growth [14]. Activation of the hypothalamic–pituitary–interrenal axis increases the circulating levels of the primary glucocorticoid in teleosts fish cortisol, which promotes mobilization and redistribution of energy and collectively retards the growth of stressed fish [15].

There is an urgent need to control aquaculture nitrogenous waste deleterious effects in order to maintain water quality, survivability and “clean” production systems. Several research studies have been conducted to mitigate these adverse impacts in fish [16,17]. Medicinal plants represent an essential part of the natural environment and an immense source of bioactive constituents [18]. These can increase fish performance, antioxidant potential, and immune responses, in addition to alleviating the stress conditions [19]. 

*Yucca schidigera* is a medicinal herb which represents an important future ecofriendly supplement in livestock production [18]. *Yucca schidigera* contains bioactive components including steroidal saponins and polyphenolics such as resveratrol and yuccaols A, B, C, D, and E [20,21]. These phenolic compounds can improve fish health and welfare [22]. Numerous studies documented that *Yucca schidigera* extract (YSE) has antioxidant, growth-promoting, anti-inflammatory, immune-stimulatory, and anti-carcinogenic effects in many species such as chickens, Japanese quail, and rabbits [21,23,24]. YSE has the ability to regulate energy metabolism and hormonal activity in animals [25].

Nile tilapia (*Oreochromis niloticus* L.) is an important cultured aquaculture species throughout the world that could be used as a suitable model for studying nutrition and metabolism not only because of its rapid growth and high resistance to diseases and toxic stress [26], but also due to the availability of its whole genomic information [27]. This species also has well-developed digestive and metabolic organs, including liver, muscle, and adipose tissues. 

Keeping in mind the aforementioned information, to date, little is known about the YSE’s molecular modulatory mechanism against ammonia stress concerning tilapia growth, energy mobilization, and inflammatory response. Hence, the present investigation aimed to declare YSE modulatory effect (s) against chronic ammonia intoxication in Nile Tilapia (*Oreochromis niloticus*). 

## 2. Materials and Methods

### 2.1. Ethical Statements

The procedure of the experiment was conducted according to the Egyptian codes of ethics and approved by the Committee of Animal Ethics at the Faculty of Aquatic and Fisheries Sciences, Kafrelsheikh University, Egypt (approval number: IAACUC-KSU-3-2020).

### 2.2. Fish, Diet and Experimental Design

A total of 120 healthy male monosex Nile tilapia (*Oreochromis niloticus)* were collected from a private farm “El-Behaira Governorate, Egypt”, with an average body weight of 42.22 ± 1.25 (mean ± SD) initial weight and transported to the Faculty of Aquatic and Fisheries Sciences, Kafrelsheikh University, Egypt. Prior to the experiment, the fish were acclimated for two weeks. The fish were then divided into four groups (30 fish per group) in 12 glass aquaria, ten fish per glass aquarium (70 × 40 × 60 cm), and the fish were randomly distributed into four groups. The first group kept as a normal control group, the second group supplied with YSE in water at a dose of 8 mg/L water every two days, the third group exposed to ammonia from the beginning of the experiment for four weeks, and the fourth group was supplied with YSE and exposed to ammonia for four weeks. The experimental design is portrayed in Figure 1. The fish were fed basal diet prepared according to NRC [28] (Table 1) that was supplied twice daily (09:00 am and 03:00 pm). YSE (3% saponins, was obtained from ANOVA Pharm Company, Tanta, Egypt) was supplied in water at 8 mg/L every two days. The chemical constituents of the basal diet were confirmed following the standard methods [29]. All the fish were weighed at the beginning of the experiment and every week until the end of the experiment (4 weeks) to readjust the feed intake and visually monitor the fish health status. The wastes were syphoned on a daily basis from all aquaria and water was exchanged with de-chlorinated water, except for the ammonia groups. 

### 2.3. Water Quality

Different water parameters were measured on a daily basis throughout the experiment’s duration, including dissolved oxygen (DO), ammonia, water temperature, and pH, in all experimental groups. The DO, pH, and temperature were detected by using a dissolved oxygen meter (Oxyguard international A/S, Farum, Denmark), and a pH and temperature meter (HANNA, Smithfield, RI, USA). The total ammonia nitrogen was measured by TAN meter (HANNA, Smithfield, RI, USA).

### 2.4. Blood Sampling, Hematological and Serum Biochemical Analysis

After four weeks, nine fish/group were collected and anesthetized using 150 mg/L MS222 (Argent Laboratories, Redmond, Washington). The blood samples were collected from fish in a random order from the caudal vein and divided into two parts. One half was treated with anti-coagulant, EDTA (dipotassium salt) (1 mg/mL blood) for total and differential leukocyte counts according to [30]. The other half of the blood samples was collected in a plain Eppendorf tube without anticoagulant, then blood samples were allowed to coagulate, centrifuged at 3000 rpm/15 min at 4 °C, and divided into aliquots stored at −20 °C for measuring the serum biochemical parameters. A spectrophotometer and commercial test kits were used for biochemical measurements of transaminases (ALT and AST), LDH, amylase, lipase, TC, TG, HDL-C, total proteins, albumin, BUN. LDL-C and VLDL-C concentrations were calculated using the standard Friedewald equation [31]. Globulin concentrations (Glob) in serum samples were computed by subtracting the albumin concentration from the total quantity of proteins while taking into account the albumin to globulin ratio (A/G) according to Kaneko [32]. 

### 2.5. Evaluation of Lipid Peroxidation and Antioxidant Enzymes

A piece of liver of about 0.5 g (9 fish/group) was rapidly removed from each fish, cleaned from any exogenous materials and immediately perfused with cold physiological saline (NaCl 0.9%). The tissues were then homogenized in cold phosphate buffer saline (0.1 M pH 7.4); then, the homogenate was filtered and centrifuged at 1500 rpm for 20 min at 4 °C using a high-speed cooling centrifuge (Type 3K-30, Sigma, Osterode-am-Harz, Germany) to separate the nuclear debris. Finally, the resulting supernatants were subjected to biochemical determination of malondialdehyde (MDA) content according to [33], and glutathione peroxidase (GPx) using the techniques outlined by [34]. Superoxide dismutase (SOD) was evaluated according to [35] and reduced glutathione (GSH) according to [36]. All tests were obtained from Biodiagnostic Co. (Dokki, Cairo, Egypt) and performed following the manufacturer’s instructions. 

### 2.6. Histopathology Study

Sections from liver, spleen, brain and gills were fixed in 10% neutral buffered formalin for at least 24 h. Tissue samples were dehydrated using ascending concentrations of ethanol (70–100%), cleared in xylene, and embedded in paraffin wax. Tissue sections of 5 μm thickness were cut on a microtome (Leica RM 2125, Wetzlar, Germany), stained with hematoxylin and eosin (H&E) and examined with a light microscope (Leica DM 5000, Wetzlar, Germany). The histopathological examination of the slides was performed under a microscope (Zeiss, Oberkochen, Germany) with ×200 magnification power [37].

### 2.7. Total RNA Extraction, cDNA Synthesis and Real-Time Quantitative PCR Assay

This work was performed in the Biotechnology Lab, Faculty of Aquatic and Fisheries Sciences, Kafrelsheikh University. To evaluate gene expression levels, liver, adipose, muscle and brain tissues (3 fish/replicate) were collected from all tested groups in 2-mL sterile Eppendorf tubes and immediately shocked in liquid nitrogen for RNA extraction. Total RNA was extracted from 50 mg tissue using Trizol (iNtRON Biotechnology) following the manufacturer’s instructions.

The integrity of extracted RNA was confirmed by ethidium bromide-stained 2% agrose gel electrophoresis. RNA concentration was assessed by NanoDrop^®^ BioDrop Spectrophotometer. Two μg of extracted RNA sample was reverse transcribed using Maxime RT PreMix (Oligo dT primer) (iNtRON Biotechnology, Seongnam-Si, Korea) following the manufacturer’s manual. Gene expression analysis was performed in the mic-PCR Real-time PCR system (Bio-molecular systems) using the SensiFast SYBR No-Rox kit (Bioline) using Nile tilapia gene-specific primers (Table 2) [38,39,40,41,42,43,44,45] with 16s rRNA as the housekeeping gene. Relative expression of selected genes was calculated by ΔΔCT method according to [46].

### 2.8. Statistical Analyses of Data

All the data were expressed as means ± SEM. Statistical analysis of the data was performed using the one-way ANOVA test followed by Tukey’s post hoc test to determine the difference between the mean values of the different groups. All statistical analyses were performed using GraphPad PRISM software (Version 8.0.2, La Jolla, CA, USA). *p* value < 0.05 was considered to indicate statistically significant differences. All data were expressed as Means ± SEM.

## 3. Results

### 3.1. Water Quality

Table 3 shows the results for measurement of parameters of the water quality including the water temperature, pH, DO (dissolved oxygen), TAN (total ammonia nitrogen) and UIA (unionized ammonia), which did not differ among the control group and YSE-supplied group (*p* > 0.05). Moreover, the pH, and the levels of TAN and UIA were significantly (*p* < 0.05) increased in the fish group exposed to elevated ammonia level when compared to the control group. However, YSE supplementation significantly reduced (*p* < 0.05) pH, TAN and UIA levels compared to the fish group exposed to high ammonia levels.

### 3.2. Growth Performance

Figure 2 portrays the growth performance of the examined fish groups. The FBW, BWG, and SGR were not significantly changed in the fish group supplied with YSE compared to the control group (*p* > 0.05). However, compared to the control group, fish exposed only to high ammonia level showed significantly lowered FBW (final body weight), BWG (body weight gain), and SGR (specific growth rate), and higher FCR (feed conversion ratio). Conversely, supplying YSE to ammonia-intoxicated fish successfully enhanced FBW, BWG, and SGR and significantly lowered FCR in comparison to the fish group exposed only to high ammonia levels.

### 3.3. Leukogram and Serum Biochemical Findings

Table 4 represents the effects of high levels of ammonia and/or the ameliorating role of YSE in terms of leukogram findings and serum biochemical analysis. The second fish group, exposed to high ammonia levels, exhibited signs of stress leukogram, which were represented by a significant (*p* < 0.05) increase in WBCs, heterophil, and monocyte counts, with a decline in lymphocyte and eosinophil counts, compared to the normal control group. However, YSE administration to ammonia-intoxicated fish restored the values of the stress leukogram to normal reference levels compared to the second fish group, which was intoxicated by high ammonia levels, where no changes occurred. 

Moreover, ammonia intoxication significantly (*p* ≤ 0.05) increased the values of serum hepatic biomarkers ALT, AST and LDH enzyme activities, and renal injury markers such as BUN with increased glucose levels, while declines occurred in serum parameters such as total proteins (TP), albumin, TC, TG, HDL-C, and VLDL-C, amylase and lipase. Conversely, YSE resulted in an effective improvement according to the measurement of ammonia-altered hepatorenal injury markers. Notably, there were no significant alterations in serum biomarkers in fish supplied with YSE alone when compared to the control fish group, indicating the safety of YSE at the selected dose used in the current study. 

### 3.4. Hepatic Lipid Peroxidation and Antioxidant Biomarkers

Figure 3 elucidates the impacts of the elevated ammonia level as well as the beneficial effects of YSE on the elucidation of hepatic lipid peroxidation and antioxidant parameters. The fish group exposed to high ammonia levels revealed a significant (*p* ≤ 0.05) increase in hepatic MDA level with a significant (*p* ≤ 0.05) depletion of hepatic GPx, SOD enzyme activities and GSH contents compared with the control fish group. However, the findings of ammonia-intoxicated fish additionally supplied with YSE were opposed to the findings for the ammonia-intoxicated group, in which there was an inhibition of hepatic MDA content and an enhancement of SOD, GPx enzyme activities and GSH levels.

### 3.5. Histopathological Observations

Figure 4 shows that ammonia intoxication induced marked degenerative and necrotic changes within the hepatopancreas (P) with a moderate degree of hepatic vacuolation (Figure 4B). However, YSE treatment markedly improved pathological changes in the hepatopancreas that were induced by high ammonia levels (Figure 4C). Moreover, melanomacrophage centers (MMC) in which macrophage aggregates contain distinctive groupings of pigment-secreting cells within the tissues of heterothermic vertebrates were markedly depleted in the ammonia-intoxicated group (Figure 4F). 

On the other hand, YSE treatment to ammonia-intoxicated fish had improved the MMC (Figure 4G). Moreover, brain sections in the second group showed a wide area of malacia associated with marked gliosis (Figure 4J). However, the administration of YSE to ammonia-intoxicated fish resulted in tiny foci of malacia with a marked decrease in gliosis (Figure 4K). Additionally, an elevated ammonia level induced severe loss of secondary lamellae with marked infiltration of inflammatory cells (Figure 4O), while YSE treatment resulted in a marked decrease in the adhesion between the secondary lamellae (Figure 4N).

### 3.6. mRNA Expression Profile

#### 3.6.1. Relative Gene Expression of Appetite- and Growth-Related Genes

Figure 5 represents the effects of high ammonia levels and/or YSE on the relative mRNA levels of the brain *NPY*, liver *IGF-1*, *MSTN* and brain *MSTN*. Compared with the control group, the second fish group, which was intoxicated by ammonia, had reduced levels of both brain NPY (Figure 5A1,A2) and hepatic *IGF1* (Figure 5B1,B2) after two and four weeks, respectively, and brain *MSTN* after two weeks only (Figure 5D1), and elevated *MSTN* gene expression levels were found in in both the liver (Figure 5C1,C2) and the brain (Figure 5D2) after four weeks. On the other hand, the fish group supplied with YSE and intoxicated with ammonia (3rd group) exhibited higher expression levels of brain NPY (Figure 5A1,A2) and hepatic IGF1 (Figure 5B1,B2) with a decreased expression of liver *MSTN* (Figure 5C1,C2) after two and four weeks, and brain *MSTN* (Figure 5D2) after four weeks, compared with the chronically intoxicated ammonia group. 

#### 3.6.2. Relative Gene Expression of Some Glucose and Lipid Regulatory Genes

Figure 6 displays the impacts of ammonia intoxication and/or YSE on relative mRNA expression levels of hepatic glycolytic enzyme PK, gluconeogenesis enzymes G6pase and PEPCK, and adipose FAS and LPL. Compared to the control group, the ammonia-intoxicated group exhibited reduced expression levels of glycolytic gene PK (Figure 6A1,A2) and adipose tissue lipogenesis encoding of the *FAS* gene (Figure 6D1,D2), as well as elevated gluconeogenesis genes *G6pase* (Figure 6B1,B2) and PEPCK (Figure 6C1,C2) and enhanced adipose tissue lipolysis regulating the *LPL* gene (Figure 6E1,E2) at two and four weeks. Interestingly, treating fish with YSE either intoxicated with ammonia or supplied alone counteracted what was expressed in the ammonia-intoxicated-only fish group at two and four weeks.

#### 3.6.3. Relative Gene Expression of Some Inflammatory Cytokine Genes

Figure 7 illustrates the findings of chronic ammonia intoxication and/or YSE treatment on relative mRNA expression levels of Nile tilapia hepatic *IL-1β*, *TNF-α* and *HSP70* genes. Compared with the control fish group, the fish group exposed to ammonia intoxication exhibited higher expression levels of hepatic *IL1β* (Figure 7A1,A2), *TNFα* (Figure 7B1,B2) and *HSP70* (Figure 7C1,C2) after two and four weeks. However, the groups supplied with YSE, either with ammonia intoxication or without, experienced down regulation of the hepatic *IL1B*, *TNFα* and *HSP70* gene expression levels compared to the ammonia-intoxicated-only fish group after two and four weeks (Figure 7A–C).

## 4. Discussion

Ammonia is a problematic environmental toxicant for aquatic species. The research regarding the ameliorative mechanism of *Yucca schidigera* extract against chronic ammonia toxicity in fish is still limited. The current study aimed to declare the modulatory effect (s) of *Y. schidigera* extract against chronic ammonia intoxication in Nile tilapia through a series of indices related to growth performance, haemato-biochemical and antioxidant-related parameters, and histopathological changes; in addition, the molecular gene expression of some genes related to appetite and growth, the glucose and lipid metabolism and some inflammatory cytokine were analyzed.

ROS induces deleterious impacts as a consequence of an imbalance between the production and inactivation of these species causing irregularities in cellular physiology and different pathological conditions [47]. In this work, stressed fish with a high ammonia level revealed a significantly elevated TAN and UIA in the water compared to the control fish group, and such results may be attributed to adverse effects of ammonia on the osmoregulation, respiration, and protein metabolism of tilapia, in addition to oxidative stress caused by the free radical production and the subsequent depletion of intracellular antioxidant enzymes [48]. Here, we detected that high ammonia level was harmful to Nile tilapia through the induction of oxidative stress via an elevated hepatic MDA content with depletion in the measured level of hepatic antioxidant defense markers.

Furthermore, the fish group exposed to a high ammonia level displayed reduced growth performance via decreased FBW, BWG and SGR, with a significant increase in FCR together with reduced serum amylase and lipase as compared with the normal control fish group. Digestive enzyme activities are good indicators of digestive capacity and precisely reflected the nutritional status of fish, and they are also highly sensitive to ROS [49]. Our results were in line with those of Xu [50], who described the induction of digestive enzyme activities at low ammonia concentrations and reported inhibition of their activities at higher levels of ammonia in *Charybdis japonica*. 

Moreover, in the present study, we found that the growth-suppressing effects of chronic ammonia exposure may have resulted from the downregulation of brain *NPY* and hepatic *ILGF-1* and the elevation of myostatin (*MYSN*) expression levels, in addition to the depletion of lipid reserves through the enhancement of adipose tissue lipolysis and the diminution of the adipo-lipogenic potential. The results of the current work are in agreement with Stern et al. [51] and Dasarathy and Hatzoglou [52], who linked increased ammonia levels with elevated *MSTN* expression in mammals. *MSTN* is a negative regulator of skeletal muscle growth that is highly conserved across vertebrate species, indicating an essential function in regulating muscular development and growth [53,54,55]. The similarities of *MSTN* functioning between mammals and fish could potentially support the role of *MSTN* concerning fish growth in ammonia toxicity. In this work, fish intoxicated with ammonia compared with control group revealed reduced *NPYa* expression with elevated serum glucose levels and enhanced gluconeogenesis and lipolytic activity of LPL, suggesting that chronic ammonia stress could indirectly inhibit food intake through the hyperglycemic and lipolytic effects. 

At the same time, the findings of [56,57] suggested high levels of serum glucose and fatty acid inhibited food intake in rainbow trout, and there is clear evidence that hypothalamic glucosensing as well as fatty acid sensing systems are involved in regulating the expression of appetite-regulating genes in this species. In terms of explaining the growth performance, the growth-related and lipid regulatory gene expression results in the current study. Numerous studies declared that chronic stress is associated with elevated plasma cortisol, which resulted in reduced feeding in channel catfish (*Ictalurus punctatus*) [58], rainbow trout (*Oncorhynchus mykiss*) [59], and sea bass (*Dicentrarchus labrax*) [60]. 

The feed conversion efficiencies were lowered by impairing nutrient absorption in the gut, in addition to increasing metabolic rates as well as mobilizing fuel reserves and inhibiting the growth-promoting effects of the GH/IGF axis [60,61]. Interestingly, and more specifically, we demonstrated here for the first time that YSE supplementation counteracted the suppressive effects of high ammonia levels on Nile tilapia growth at the molecular level through enhancement of brain *NPYa*, hepatic *ILGF1* and inhibition of hepatic *MSTN* expression levels. Our results are consistent with Alagawany et al. [21], Chrenkova et al. [62], Aazami et al. [63], Cheeke at al. [64] and Wang et al. [65] who implied that yucca can enhance the variety and viability of advantageous microorganisms by facilitating the nutrients’ digestibility in addition to its effect in enhancing the digestive enzyme activities towards providing better growth, production performance and immune responses. Our results are in line with those of Gaber [66] and Njagi et al. [67] who documented that Nile tilapia that were fed dietary yucca displayed enhanced growth performance, which may be attributed to yucca’s polyphenolic and phytochemicals compounds (e.g., resveratrol and yuccaols A, B, C, D, and E), which act as natural growth promoters and antioxidants and as a good substitute for both antibiotics and ionophores [20,68].

The hemato-biochemical indices reflect hepatic, renal and immune functions, as well as lipid and protein metabolism, in the blood of the organisms [69]. Therefore, the general physiological and health condition of fish reared under stressful conditions or specifically supplied with a modulatory agent can be well characterized [70]. In this work, we challenged fish with high ammonia level whose leukogram picture was indicative of stress.

The results of the recent work are consistent with those of Grzelak et al. [71] who detected leukocytosis with significant lymphopenia, monocytosis and neutrophilia in acutely stressed Zebra fish (*Danio rerio*) compared with unstressed fish. Stress conditions caused cortisol secretion that curtailed lymphocytes life span and induced their apoptosis, thus producing a decline in lymphocyte numbers [70]. The results of Amulic et al. [72] supported the association between the observed neutrophilia, elevated hepatic MDA content and depletion of hepatic antioxidant activities through numerous inflammatory mediators, and this is consistent with our results. Similarly, Jaeschke and Hasegawa [73] stated that neutrophils induced oxidative DNA damage. The overproduction of ROS initiates inflammation [74]. Oxidative stress alters several cellular functions via numerous mechanisms, such as alteration in inflammatory gene expression [75]. In the present study, we found marked up-regulation in hepatic gene expression levels of pro-inflammatory cytokines *TNFα* and *1L1**Ɓ* in ammonia-intoxicated fish compared to the control fish group. Interleukins (*ILs*) and tumor necrosis factor α (*TNFα*) are crucial pro-inflammatory cytokines which play a significant role in the fish immune response that contributes to the host defense by initiating and exaggerating inflammatory processes [76].

The results are in line with [74] who stated that excessive production of ROS promoted the production of *TNF-α* and *IL-1**Ɓ* pro-inflammatory cytokines, which can lead to cell apoptosis and the initiation of inflammation. Furthermore, in this study, we described several types of organ damage, as confirmed in our histopathological examinations, that were in line with the findings of Benli et al. [10], who detected cloudy swelling and hydropic degenerations in Nile tilapia liver exposed to sub-lethal concentrations of ammonia. Similarly, Thurston et al. [77] detected liver glycogen vacuolation due to disruption of energy production after ammonia exposure.

Regarding the adverse effects of ammonia, Wajsbrotet al. [7] observed clear signs of liver pathology in gilthead sea bream (*Sparus auratus*) 20 days after exposure to 13 mg/L TAN. Our histopathological findings were supported by the release of serum hepato-renal injury biomarkers such as ALT, AST, LDH and urea, which may be attributed to direct hepatic and renal injuries induced by ammonia oxidative damage, which enhanced lipid peroxidation and exhausted the antioxidant defense enzymes, consequently leading to tissue injury, necrosis and the release of intracellular enzymes including ALT, AST and LDH.

In the same context, Cheng et al. [78] claimed that ammonia intoxication induced ROS production, and that the increased MDA reflects the inability of the antioxidant defense system to counteract the ROS-induced tissue injuries. Fayed et al. [48] reported elevated ALT, AST and LDH activities, as well as BUN and glucose levels as a sequel in ammonia’s adverse impact on tilapia. Interestingly, YSE supplementation for ammonia-intoxicated fish succeeded in producing a decline in the hepatic expression levels of *TNF-α* and *IL-1**Ɓ* and a stress alteration in *HSP**70*.

Farag et al. [23] found that quail with a diet treated with YSE demonstrated a significant decline in serum levels of TNF-α, which was related to its ability to decrease pro-inflammatory cytokine levels. In addition to counteracting the oxidative stress and maintaining a balanced cellular redox state, this enhanced the cells involved in the antioxidant defense systems, allowing them to prevent or repair such damage. Gupta [79] and Alagawany et al. [21] attributed the modulating role of YSE against the adverse effects of ammonia stress to its potent antioxidant, anti-inflammatory and immunostimulatory properties via saponin and phenolic contents.

In the same way, Alagawany et al. [80] attributed the antioxidant activities of yucca to its polyphenolic compounds (resveratrol (RES) and yuccaols A, B, C, D, and E) and steroidal saponins. Besides, Olas et al. [81] stated that yucca can inhibit free radical generation, and thus reduce lipid peroxidation in blood platelets. As yucca phenolic hydroxyl groups act as hydrogen donors for the proxy radicals produced in the first stage of lipid oxidation, this lowered and inhibited the formation of hydroxyl peroxide [80], thus playing an important role in its protective effect.

In the current work, ammonia intoxication was observed to lower the serum total proteins and albumin concentrations compared to the control fish group, which indicates the severe impairment of the tilapia immune system. These findings are consistent with those of Elbialy et al. [70], who claimed that the stressful conditions favored reactive oxygen species (ROS) production in stressed fish livers and, therefore, induced oxidative stress, which damaged the hepatocytes, followed by the subsequent impairment of hepatocytes’ function and protein metabolism. However, high ammonia-stressed tilapia supplied with YSE demonstrated enhanced TP and albumin concentrations, reflecting the impact of YSE in regulating the nutrient metabolism of tilapia reared under either optimal or stressful conditions [68]

Regarding the serum lipid profile findings, we detected that the fish group exposed to high ammonia levels demonstrated reduced levels of TC, TG, HDL-C and VLDL-C, with raised levels of LDL-C compared to the control fish group. Gaber [66] detected a significantly higher body protein content and lowered body lipid content in Nile tilapia whose diets were supplied with yucca compared with fish fed a control diet. However, research on lipid metabolism in ammonia-intoxicated Nile tilapia, particularly at the molecular level, is scarce.

Fatty acid synthase (FAS) is a key enzyme in the biosynthesis process of long-chain fatty acids; however, inhibiting this enzyme reduces lipid deposition [82,83]. In this study, we found that ammonia-intoxicated fish showed a reduced adipose tissue *FAS* expression level and enhanced *LPL* expression level compared to the control fish group. These results are in agreement with those published by the authors of [84,85], who reported that ammonia exposure caused alterations in the lipid metabolism of chickens, and the findings of [86,87], who demonstrated this in aquatic animals. The results from the current study provided a novel perspective on the multiple interacting pathways through which ammonia stress contributes to the coordination of energy mobilization and redistribution in Nile tilapia.

Furthermore, Madison et al. [88] stated that cortisol enhanced carbohydrate and lipid catabolism during stress. The impact of chronic stress on energy reserves can be explained by a depression in liver lipid content, which is in agreement with the elevated lipolytic capacity of chronically stressed fish [89]. The enhanced activity of various lipases by cortisol can diminish lipid reverses in fish [90]. The liver provides a vital role in regulating the whole-body metabolism in accordance with nutritional status, regarding glucose storage and glucose production, which are important for maintaining glucose homeostasis through regulation of the activity and gene expression of key enzymes involved in glycolysis, glycogenolysis, lipogenesis and gluconeogenesis [91].

During chronic stress, cortisol influences the liver physiology of fish by modulating the carbohydrate- and lipid-related metabolism [92]. In this study, the fish group that was intoxicated with ammonia exhibited elevated serum glucose level, and also showed higher ALT, AST and LDH enzyme activities, with downregulation of the hepatic gene expression of *PK* and an increase in *G6Pase* and *PEPCK* gene expression levels, compared to the control fish group. Our data reinforced the hypothesis that the absence of a concomitant increase in hepatic glycolysis and the stimulation of gluconeogenesis in the liver may contribute to the establishment of elevated blood glucose levels.

Nayanatara et al. [93] and Nagaraja et al. [94] stated that stress increased the activities of aminotransferases, which further increased the concentration of substrates such as pyruvate and oxaloacetate for gluconeogenesis. Cortisol enhances alanine oxidation and alanine gluconeogenesis [95]. Moreover, Wang [96] documented that blood glucose levels not only increased with gluconeogenesis, but also when reducing insulin sensitivity through the antagonizing of the insulin-stimulated translocation of glucose transporters from intracellular compartments to the plasma membrane during stress caused by glucocorticoids.

Nirupama [13] stated that chronic hyperglycemia during stress enhanced the endogenous synthesis of glucose. Numerous fish studies have reported an enhancement of hepatic gluconeogenic capacity and an increase in the plasma glucose levels caused by cortisol [97,98]. Finally, the above findings are in agreement with those of Marandel et al. [99] and Song et al. [100], who indicated that glucose-intolerance (hyperglycemia) is due to the reduction in glycolysis and the elevation of gluconeogenesis in livers of rainbow trout.

## 5. Conclusions

Our results provided a novel perspective on the multiple interacting mechanisms through which YSE may exert its protective role against chronic ammonia toxicity. These results affirmed the growth-enhancing effects of YSE via the sustained enhancement of food intake, the elevation of *IGF-1*, the suppression of hepatic and brain *MTSN* expression levels, and the restoration of carbohydrate and lipid reserves, mediated through alterations in the levels of circulating metabolites.

Our results indicated that *Yucca schidigera* extract alleviated the adverse impacts induced by ammonia intoxication through its ability to scavenge free radicals, potent antioxidant activities and anti-inflammatory properties. The results of this study suggested that YSE supplementation was clearly beneficial for both health and growth in Nile tilapia, and that YSE could be used as a functional water supplement in aquaculture.

## Figures and Tables

**Figure 1 animals-11-02072-f001:**
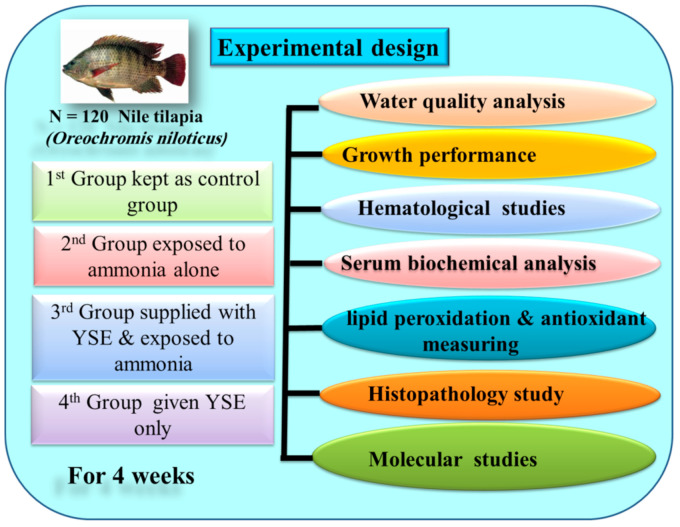
Experimental design showing different treatment groups.

**Figure 2 animals-11-02072-f002:**
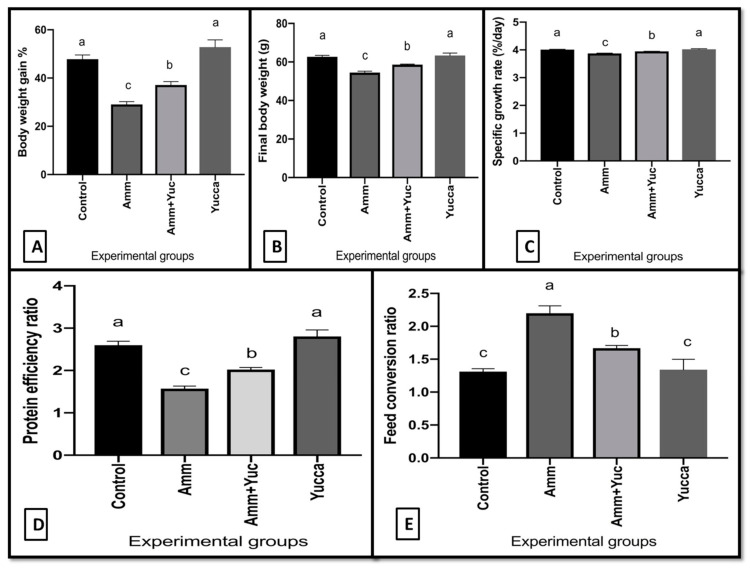
Growth performance parameters of control Nile tilapia and yucca supplied groups either alone or stressed by high ammonia level for 30 days. (**A**) Body weight gain %, (**B**) Final body weight, (**C**) Specific growth rate, (**D**) Protein efficiency ratio, (**E**) Feed conversion ratio. Values are expressed as mean ± SE. Bars with different letters are significantly different from those of control group (*p* < 0.05). *n* = 9.

**Figure 3 animals-11-02072-f003:**
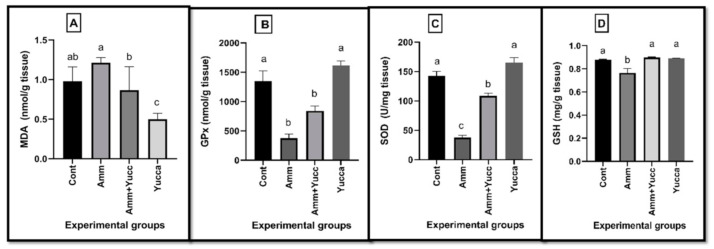
(**A**) Malondialdehyde: MDA; (**B**) Glutathione peroxidase: GPx; (**C**) superoxide dismutase: SOD; (**D**) Reduced glutathione: GSH concentration of control Nile tilapia yucca supplied groups either alone or stressed by high ammonia level for 30 days. Values are expressed as mean ± SE. Bars with different letters are significantly different from those of control group (*p* < 0.05). *n* = 9.

**Figure 4 animals-11-02072-f004:**
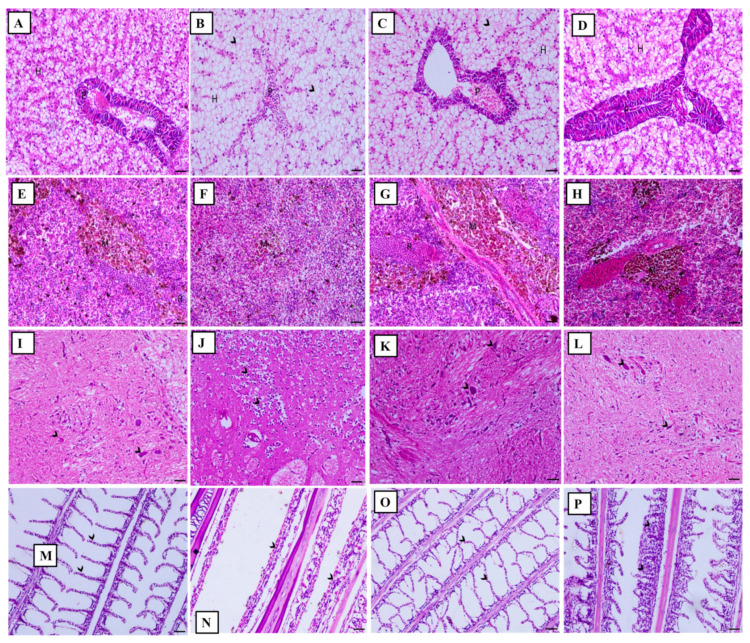
(**A**) Liver section of normal fish showing normal hepatic cells (H) and hepatopancreas (P). (**B**) Liver section of high ammonia level stressed fish showing marked degenerative and necrotic changes within the hepatopancreas (P) and moderate degree of hepatic vacuolation (H indicates hepatocytes). (**C**) Liver section of fish supplied with yucca extract and stressed by high ammonia level showing marked decrease the degenerative changes within the hepatocytes (H) and the hepatopancreas (P). (**D**) Liver section of fish treated with yucca extract showing normal hepatic cells (H) and hepatopancreas (P). (**E**) Spleen section of normal fish showing normal white pulp containing the melanomacrophage centers (M) and red pulp containing the blood capillaries (R). (**F**) Spleen section of high ammonia level stressed fish showing marked depletion of melanomacrophage centers (M). (**G**) Spleen section of fish supplied with yucca extract and stressed by high ammonia level showing improvement in the melanomacrophage centers (M). (**H**) Spleen section of fish treated with yucca extract showing normal melanomacrophage centers (M) and red pulp (R). (**I**) Brain section of telencephalon portion of normal control fish showing normal neuronal cell (arrowhead) and nerve fibers. (**J**) Brain section of telencephalon portion of fish stressed by high ammonia level showing wide area of malacia (arrows) associated with marked gliosis (arrowheads). (**K**) Brain section of telencephalon portion of fish stressed by high ammonia level and supplied with yucca extract showing tiny foci of malacia (arrow) and with marked decrease in gliosis (arrowheads). (**L**) Brain section of telencephalon portion of fish treated with yucca extract showing normal neuronal cell (arrowheads). (**M**) Gills section of normal fish showing normal secondary lamellae (arrowheads). (**N**) Gills section of fish stressed by high ammonia level showing severe loss of secondary lamellae associated with marked infiltration of inflammatory cells (arrowheads). (**O**) Gills section of fish stressed by high ammonia level and supplied with yucca extract showing marked decrease in the adhesion between the secondary lamellae (arrowheads). (**P**) Gills section of normal fish treated with yucca extract showing normal secondary lamellae (arrowheads). H&E, X200 bar = 50 µm. *n* = 9.

**Figure 5 animals-11-02072-f005:**
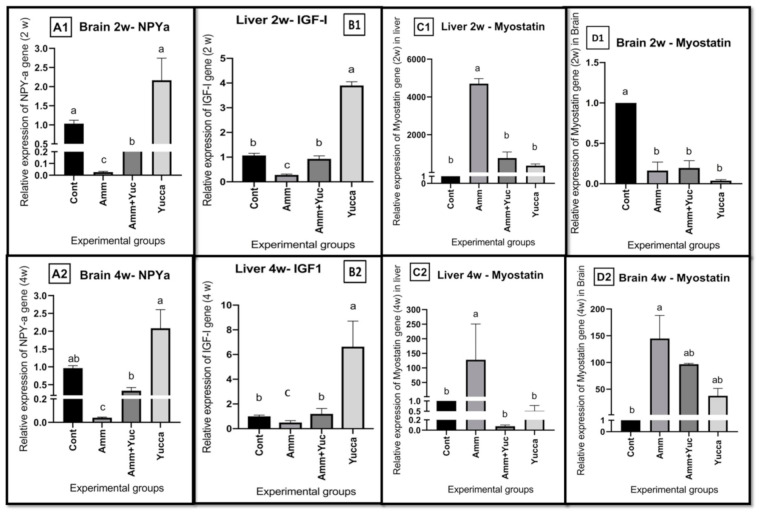
The effects of high ammonia level and/or YSE on the relative mRNA levels of (**A1**,**A2**) brain Neuropeptide Y: NPY; (**B1**,**B2**) Hepatic Insulin like growth factor1: IGF-1; (**C1**,**C2**) liver myostatin (MSTN) and (**D1**,**D2**) brain myostatin (MSTN) after 2 and 4 weeks, respectively, of Nile tilapia for 30 days. Different letters refer to significantly different effects in the same time period. Means ± SEM. *n* = 9.

**Figure 6 animals-11-02072-f006:**
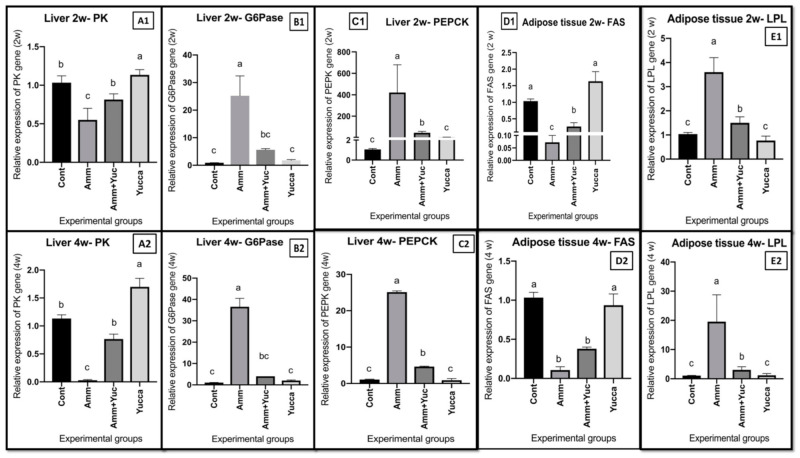
The effects of high ammonia level and/or YSE on the relative mRNA levels of (**A1**,**A2**) hepatic pyruvate kinase: PK; (**B1**,**B2**) Hepatic Glucose 6 phosphatase: G6Pase; (**C1**,**C2**) hepatic phoshoenolpyruvate carboxykinase: PEPCK; (**D1**,**D2**) adipose tissue fatty acid synthase: (FAS) and (**E1**,**E2**) adipose tissue lipoprotein lipase (LPL) after 2 and 4 weeks, respectively, of Nile tilapia for 30 days. Different letters refer to significantly different effects in the same time period. Means ± SEM. *n* = 9.

**Figure 7 animals-11-02072-f007:**
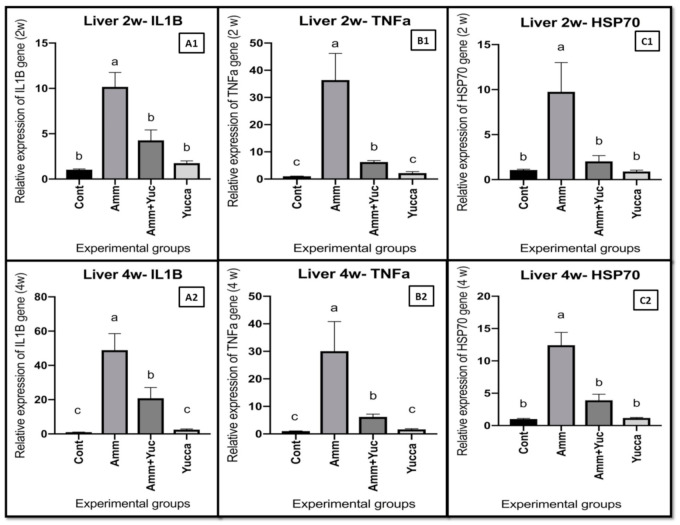
The effects of high ammonia level and/or YSE on the relative mRNA levels of Hepatic (**A1**,**A2**) Interleukin 1B; (**B1**,**B2**) Tumor necrosis actor alpha: TNFα and (**C1**,**C2**); heat shock protein 70: HSP70 after 2 and 4 weeks, respectively, of Nile tilapia for 30 days. Different letters refer to significantly different effects in the same time period. Means ± SEM. *n* = 9.

**Table 1 animals-11-02072-t001:** Formulation and chemical composition of the basal diet.

Ingredients	%	Chemical Composition	%
Fish meal (62%)	5.80	Crude protein (%)	30.08
Soybean meal (48%)	38.00	Lipid (%)	6.30
Wheat middling	6.80	Fibre (%)	5.10
Corn (7.5% CP)	28.25	Ash (%)	6.01
Corn Gluten (63.1)	6.00	Gross energy (K.cal/kg)	4180.8
Fish oil	1.40		
Dicalcium Phosphate	0.30		
Rice bran	13.10		
Vitamin C (35%)	0.05		
Vitamin and mineral mix	0.30		

After 4 weeks, all fish in the experimental groups were weighed to calculate the growth performance parameters as shown in the following equations: Weight gain (%) = (W1 − W2) × 100/W1. Specific growth rate (% per day) = {(Ln (W2) − Ln (W1))/duration (4 weeks)} × 100. Feed conversion ratio = dry feed intake (g)/live weight gain (g). W1 and W2 are initial and final body weight (g) of the fish.

**Table 2 animals-11-02072-t002:** Primers sequences used for Q-rtPCR.

Gene	Primer Sequence 5′-3′	NCBI Accession Number	Reference
*18s rRNA* *	F: GGACACGGAAAGGATTGACAG R: GTTCGTTATCGGAATTAACCAGAC	JF698683	[38]
*G6Pase*	F:AGCGCGAGCCTGAAGAAGTACT R:ATGGTCCACAGCAGGTCCACAT	XM_003448671	[39]
*PEPCK*	F: GCCCTCAGTCCAGCTGTAAG R: CACATCCCTCGGGTCAGTTC	XM_003448375.4	This study
*PK*	F:CCGTAAGGCTGCAGACGTGCA R: ATCTGCGCACGCCCTCATGG	DQ066876.1	[40]
*IGF1*	For: TCCTGTAGCCACACCCTCTC R: ACAGCTTTGGAAGCAGCACT	NM_001279503.1	[41]
*HSP70*	F: CATCGCCTACGGTCTGGACAA R: TGCCGTCTTCAATGGTCAGGAT	FJ207463.1	[42]
*IL1β*	F: CAAGGATGACGACAAGCCAACC R: AGCGGACAGACATGAGAGTGC	XM_003460625.2	[42]
*TNFα*	F: GGAAGCAGCTCCACTCTGATGA R: CACAGCGTGTCTCCTTCGTTCA	JF957373.1	[42]
*FAS*	F: TGAAACTGAAGCCTTGTGTGCC R: TCCCTGTGAGCGGAGGTGATTA	GU433188.1	[43]
*LPL*	F: TGCTAATGTGATTGTGGTGGAC R: GCTGATTTTGTGGTTGGTAAGG	GU433189.1	[43]
*NPYa*	F: ACAAGACAGAGGTATGGGAAGA R: GGCAGCATCACCACATTG	XM_003448854.1	[44]
*MSTN*	F: GCATCTGTCTCAGATCGTGCT R: TGCCATCATTACAATTGTCTCCG	KT987208.1	[45]

G6Pase: Glucose 6 phosphatase; PEPCK: Phosphoenolpyruvate carboxykinase; PK: pyruvate kinase; ILGF1: Insulin like growth factor 1; HSP70: Heat shock protein 70; IL1B: Interleukin 1 beta; TNFα: Tumor necrosis actor alpha; FAS: fatty acid synthase; LPL: lipoprotein lipase; NPYa: Neuropeptides a; MSTN: Myostatin; * housekeeping gene.

**Table 3 animals-11-02072-t003:** The water characteristics under the trial conditions.

	Control	Ammonia	Ammonia + Yucca	Yucca	*p* Value
Dissolved oxygen (mg/L)	5.59 ± 0.41	5.16 ± 0.43	4.92 ± 0.16	5.64 ± 0.76	0.322
Temperature (°C)	26.59 ± 0.42	27.2 ± 0.8	26.9 ± 0.72	26.80 ± 0.59	0.381
pH	7.94 ± 0.21	8.18 ± 0.11	8.03 ± 0.17	7.81 ± 0.27	0.419
TAN (mg/L)	0.37 ± 0.09 ^c^	4.1 ± 0.03 ^a^	1.41 ± 0.02 ^b^	0.41 ± 0.12 ^c^	0.001
UIA (mg/L)	0.0194 ± 0.006 ^c^	0.44 ± 0.02 ^a^	0.124 ± 0.04 ^b^	0.016 ± 0.001 ^c^	0.0017

TAN: Total ammonia nitrogen concentration; UIA: un-ionized ammonia (mean ± SEM). Different superscripts refer to differences between all groups for each parameter (*p* < 0.05). *n* = 9.

**Table 4 animals-11-02072-t004:** Leukogram and serum biochemical findings of different groups in the experiment.

	Control	Ammonia	Ammonia + Yucca	Yucca	*p*-Value
WBCs (×10^3^/µL)	16.77 ± 0.84 ^b,c^	23.84 ± 1.42 ^a^	19.65 ± 0.85 ^b^	18.69 ± 0.45 ^c^	0.0016
Neutrophils (×10^3^/µL)	6.1 ± 0.33 ^c^	14.19 ± 0.94 ^a^	8.87 ± 0.48 ^b^	7.3 ± 0.84 ^c^	0.0001
Lymphocytes (×10^3^/µL)	9.9 ± 1.76 ^b^	7.87 ± 0.317 ^c^	9.56 ± 0.22 ^b^	10.5 ± 1.61 ^a^	0.0117
Monocytes (×10^3^/µL)	0.564 ± 0.05 ^c^	1.63 ± 0.24 ^a^	0.86 ± 0.119 ^b^	0.689 ± 0.03 ^c^	0.005
Eosinophils (×10^3^/µL)	0.208 ± 0.05 ^a^	0.15 ± 0.03 ^b^	0.272 ± 0.27 ^a^	0.206 ± 0.02 ^a^	0.002
ALT (U/L)	14.33 ± 3.85 ^c^	64.21 ± 3.84 ^a^	26.67 ± 6.8 ^b^	14.1 ± 2.6 ^c^	0.0001
AST (U/L)	48.67 ± 4.49 ^c^	105.7 ± 8.09 ^a^	63.01 ± 12.06 ^b^	44.3 ± 6.69 ^c^	0.005
LDH (U/L)	457 ± 40.22 ^c^	914.5 ± 50.22 ^a^	580.4 ± 34.2 ^b^	487 ± 0.34 ^c^	0.001
Total Protein (g/dL)	4.36 ± 0.33 ^a^	3.08 ± 0.17 ^c^	4.1 ± 0.05 ^b^	4. 79 ± 0.15 ^a^	0.003
Albumin (g/dL)	2.6 ± 0.15 ^a^	1.7 ± 0.25 ^b^	2.26 ± 0.23 ^a^	2.5 ± 0.06 ^a^	0.016
Globulin (g/dL)	1.76 ± 0.19	1.38 ± 0.24	1.84 ± 2.08	2.29 ± 0.17	0.669
A/G ratio	1.48 ± 0.24	1.23 ± 0.34	1.23 ± 0.54	1.09 ± 0.24	0.4944
Cholesterol (mg/dL)	130.67 ± 11.79 ^a^	106.3 ± 11.7 ^c^	129.7 ± 28.9 ^b^	134.33 ± 21.17 ^a^	0.002
Triglyceride (mg/dL)	383 ± 56.63 ^a^	184 ± 38.5 ^c^	324 ± 24.17 ^b^	402 ± 187.02 ^a^	0.0017
HDL (mg/dL)	33 ± 5.29 ^a^	21.67 ± 3.28 ^b^	31.3 ± 3.12 ^a^	35.67 ± 6.94 ^a^	0.0017
LDL (mg/dL)	21.07 ± 0.86 ^c^	47.83 ± 16.48 ^a^	33.6 ± 27.74 ^b^	18.26 ± 0.77 ^c^	0.0058
VLDL (mg/dL)	76.6 ± 2.28 ^a^	36.8 ± 1.10 ^c^	64.8 ± 1.92 ^b^	80.4 ± 3.50 ^a^	0.002
Amylase (U/L)	70.5 ± 1.5 ^a^	65 ± 20 ^b^	75.5 ± 9.5 ^a^	73.1 ± 7.03 ^a^	0.005
Lipase (U/L)	35.5 ± 2.51 ^a^	18.2 ± 2.1 ^b^	30.67 ± 0.88 ^a^	30.1 ± 2.12 ^a^	0.005
Urea	14.01 ± 1.53 ^b^	42.03 ± 10.79 ^a^	20.32 ± 1.20 ^b^	16.67 ± 1.85 ^b^	0.0114
Glucose (mg/dL)	74.33 ± 2.3 ^c^	298.3 ± 41.2 ^a^	96.3 ± 5.77 ^b^	82.1 ± 3.09 ^c^	0.0015

White blood cells: WBCs, Albumin, globulin ratio: A/G, High density lipoproteins: HDL, Low density lipoproteins: LDL, Very low density lipoproteins: VLDL-C. (mean ± SEM), *n* = 9. Different superscripts refer to differences between all groups for each parameter (*p* < 0.05).

## Data Availability

The authors confirm that the data supporting the findings of this study are available within the article.

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
