# Peer review of "Exploring the Multimodal Role of Yucca schidigera Extract in Protection against Chronic Ammonia Exposure Targeting: Growth, Metabolic, Stress and Inflammatory Responses in Nile Tilapia (Oreochromis niloticus L.)"

_animals, 2021, doi:10.3390/ani11072072_

Round 1

Reviewer 1 Report

Exploring the Multimodal Role of Yucca schidigera Extract in Protection against Chronic Ammonia Exposure Targeting: Growth, Metabolic, Stress and Inflammatory Responses in Nile tilapia (Oreochromis niloticus L.): as Determined by Gene Expression

Research on the possibilities of using extracts from the Yucca schidigera plant in aquaculture has almost twenty years of history. An attempt to describe and characterize them was found in an article recently published in the journal Animals (Yucca schidigera Usage for Healthy Aquatic Animals: Potential Roles for Sustainability. Animals 2021, 11 (1), 93; https://doi.org/10.3390/ani11010093) . The authors of this article approached the issue very broadly. In their work, they collected dozens of literature on Yucca schidigera and on the description of the potential role of extracts from this plant in aquaculture. The results presented in the cited publications concern a number of commercial fish species, both freshwater and marine.

The authors of the article currently received for review describe the results of an experiment aimed at examining the role of yucca extract in the protection of Nile tilapia against chronic exposure to ammonia. The experiment was very well thought out and comprehensively conducted. The observations of fish in four experimental groups included the assessment of growth, metabolism, stress indices and inflammatory reactions in Nile tilapia (Oreochromis niloticus L.). The studies of the expression of selected genes were used to evaluate the physiological parameters.

The experiment was performed without repetitions and the data presented are the results obtained in this one particular experiment. Also the duration of the experiment (four weeks) seems too short to me. This lowers the value of the presented data.

Personally, I am critical of the use of Yucca schidigera extract in aquaculture. In this synthesis, I miss an analysis of the economic aspects of using Yucca schidiger in aquaculture.

 However, I believe that the peer-reviewed manuscript can be a valuable source of information allowing us to broaden our knowledge on the possibility of using natural methods of prevention in aquaculture. At the same time, the presented results allow for attempts to explain the mechanisms of action of yucca extract in fish. Therefore, I recommend publishing the article to the editorial office of the journal.

However, I would like to point out that while the experiment and its substantive description in the article are highly appreciated, the text of the manuscript is not prepared carefully enough and does not comply with the requirements for the authors. I have a few comments that I would like to submit for the attention of the editorial office of the journal:

- According to the information for authors, the sources used in the text of literature information are indicated by entering the number of the publication's position on the References list. In this manuscript, some of the literature sources are indicated by the item number on the References list, and some by the name of the first author and the number from the list.

- I have reservations about the References chapter. In the instructions for authors, there is an example of how to write bibliographic data for articles. It shows that References in bold only indicates the year of publication of the article. However the names of the authors have been written in bold in the reviewed text. Where should the year of publication of the quoted article be placed?

References provide abbreviated versions of the names of the journals. However, in some cases these names have dots abbreviated (eg items 1,2, ...) and in some cases they lack them (eg items 13, 15, ...). The manner of recording should be unified.

It is incomprehensible to give in items 53 and 54 the abbreviation et.al., when all the authors of the article were mentioned in the bibliographic data?

Most of the Latin names appearing in the headings of the cited articles are in normal typeface. However, there is a rule that the Latin generic and species names are written in italics (Yucca schidigera). Taking into account the above, I suggest that you carefully review the REFRENCES chapter and improve it according to the editorial requirements.

Author Response

Thank you for your valuable comments

  • The experiment was performed without repetitions and the data presented are the results obtained in this one particular experiment. Also the duration of the experiment (four weeks) seems too short to me. This lowers the value of the presented data.

The experiment was performed in triplicates, the aim of the study was to inspect the modulatory effect of YSE on ammonia stress which is considered acute stress in fish farms and need not to take long time to show its adverse effects.

  • Personally, I am critical of the use of Yucca schidigeraextract in aquaculture. In this synthesis, I miss an analysis of the economic aspects of using Yucca schidiger in aquaculture.

YSE has been used in aquaculture sector lately and a number of commercial products are available with considerable prices.

  • According to the information for authors, the sources used in the text of literature information are indicated by entering the number of the publication's position on the References list. In this manuscript, some of the literature sources are indicated by the item number on the References list, and some by the name of the first author and the number from the list.

The references in the text have been corrected

  • I have reservations about the References chapter. In the instructions for authors, there is an example of how to write bibliographic data for articles. It shows that References in bold only indicates the year of publication of the article. However the names of the authors have been written in bold in the reviewed text. Where should the year of publication of the quoted article be placed?

Reference section has been corrected according to the journal's guidelines

  • References provide abbreviated versions of the names of the journals. However, in some cases these names have dots abbreviated (eg items 1,2, ...) and in some cases they lack them (eg items 13, 15, ...). The manner of recording should be unified.

The references have been corrected.

  • It is incomprehensible to give in items 53 and 54 the abbreviation et.al., when all the authors of the article were mentioned in the bibliographic data?

Have been corrected

  • Most of the Latin names appearing in the headings of the cited articles are in normal typeface. However, there is a rule that the Latin generic and species names are written in italics (Yucca schidigera). Taking into account the above, I suggest that you carefully review the REFRENCES chapter and improve it according to the editorial requirements.

Has been corrected

Reviewer 2 Report

This study evaluates the benefits of using yuca extracts to reduce the aquaria ammonia for tilapia culture. The manuscript needs to be improved in the writing and description of methods and results. Results section must be elaborated and description and figures have to coincide. The topic is interesting and worthy of investigation.

Comments:

  • It is not known if the aquaria consisted on open circulating water or closed one. How the ammonium concentration was maintained constant? And YSE? How the water exchange was performed? In line 151 the ammonia groups were not syphoned or refilled? Please, clarify these aspects of the fish handling.
  • Line 181 has no end.
  • Table 3 lacks superscripts indicating the statistics.
  • Table and figure legends have to be auto explicative including the treatments, what is presented, n, SD/SEM, statistical analysis, abbreviations, etc.
  • In table 4, the number of neutrophils, eosinophils, lymphocytes and monocytes should be done as percentages.
  • All the graphs need to have the same group order (see fig. 3A). Some graphs should be in log scale to see the control.
  • Discussion section might be shortened.

Author Response

Thank you for your valuable comments

  • It is not known if the aquaria consisted on open circulating water or closed one.

The aquaria consisted of closed water which is changed regularly with syphoning of the wastes in some groups, except for ammonia-treated groups where water is not changed to keep the ammonia concentration.

  • How the ammonium concentration was maintained constant? And YSE?

The ammonia concentration was measured in regular basis to ensure constant level, together with keeping wastes and not changing water in the ammonia-treated groups.

For YSE, it was supplied every two days to ensure constant supply.   

  • How the water exchange was performed? In line 151 the ammonia groups were not syphoned or refilled? Please, clarify these aspects of the fish handling.

The wastes were syphoned on a daily basis from all aquaria and water was exchanged with de-chlorinated water except for ammonia groups to ensure high ammonia level in the ammonia-treated groups

  • Line 181 has no end.

Has been corrected

  • Table 3 lacks superscripts indicating the statistics.

Has been modified

  • Table and figure legends have to be auto explicative including the treatments, what is presented, n, SD/SEM, statistical analysis, abbreviations, etc.

Has been corrected.

  • In table 4, the number of neutrophils, eosinophils, lymphocytes and monocytes should be done as percentages.

The absolute number is more accurate in these measurements

  • All the graphs need to have the same group order (see fig. 3A). Some graphs should be in log scale to see the control.

Has been modified

  • Discussion section might be shortened.

The experiment contains a lot of parameters which must be all explained and discussed.

Reviewer 3 Report

I was honored to review the manuscript entitled “Exploring the multimodal role of Yucca schidigera extract in protection against chronic ammonia exposure targeting: Growth, metabolic, stress and inflammatory responses in Nile tilapia (Oreochromis niloticus L.): as determined by gene expression” submitted to Animals.  Phytobiotics and herbal extracted for increasing growth and health status is one of the major focus in fish nutrition research. Furthermore, phytotherapies are cost effective, environment friendly and more eco-friendly than synthetic molecules. The number of samples used in this study are scientifically suitable. Herein, the reviewer thought that this manuscript needs to improve in English language for preparing more readable article. The study presents good quality and valuable to use this herbal extract in aquaculture industry to increase production. However, there are some points that the authors should correct them and still needs some change for improving the manuscript.

There are some points to correct:

  • The authors should provide the simple summary according to guideline of the journal.
  • The abbreviation section should not be placed after abstract.
  • Please provide a full name of FBW, SGR, PER, FCR and etc. You should write the full name at the first time to make a easier for reader.
  • It would be better if you could briefly write about the methods that you utilized during the experiment in abstract.   
  • Line 89: Due to
  • Line 94: It would be better to write whole name of ROS not in abbreviation forms.
  • Line 111: after reference number [19], how to write number [38]?
  • Line 206: [37] check all references in the main body of the manuscript for making sure they have been written according to animals guideline.
  • RNA concentration would be better to assess with the Tapestation. This is more sensitive than nanodrop.
  • Table 2: PK R: 5'ATCTGCGCACGCCCTCATGG, What is 5'in the sequence?
  • Provide all Rev and For in the same ways half of them were written in F and R and half of them in Rev and For.
  • Please improve the quality of images in figure 4.
  • Line 354 and 359: ILGF or IGF?
  • Line 436: References should be written according to guideline.
  • Line 439:NPYa?
  • Line 447: For example, …..
  • Rewrite line 447-451
  • Line 450: The authors should concern that if you would like to write scientific name of fish, it is better to write for all of them not for one. Please write for all or just omit for sea bass.
  • Line 454: [60,61], please check in all of article.
  • Line 476: Zebrafish
  • Please rewrite the line numbers 478-481.
  • Line 486: IL-1β and TNF-α
  • Discussion section is to long with a lot of repeat words and sentences as well as this section was not well written and needs to edit.
  • Author contribution and references are not according the guideline.

In conclusion I believe, in my opinion that the present manuscript can be accepted after major revision, for publication. Moreover, I suggested authors to download the Animals guideline and prepare the manuscript according it.

Author Response

Thank you for your valuable comments

  • The authors should provide the simple summary according to guideline of the journal.

Simple summary has been added

  • The abbreviation section should not be placed after abstract.

The abbreviation section has been removed to the end of the manuscript

  • Please provide a full name of FBW, SGR, PER, FCR and etc. You should write the full name at the first time to make a easier for reader.

Have been corrected

  • It would be better if you could briefly write about the methods that you utilized during the experiment in abstract.

The abstract is limited with a word count and the experimental design has been referred to in the abstract  

  • Discussion section is to long with a lot of repeat words and sentences as well as this section was not well written and needs to edit.

The experiment contains a lot of parameters which must be all explained and discussed.

  • Author contribution and references are not according the guideline.

Has been corrected according to journal's guidelines

All typing errors have been corrected

Reviewer 4 Report

The study is logically based on known effects of Yucca schidigera extract (YSE) in other species (lines 110-117). An animal ethics statement is provided in section 2.1. The experimental design, materials and methods are clear and well described. Figure 1 effectively summarises the experimental design in a diagram. Results are presented comprehensively, including haematology, biochemistry, histopathology and gene expression. The results presented support use of Yucca schidigera (YSE) extract to counteract ammonia toxicity in aquaculture. Comments Whilst a list of abbreviations is provided after the abstract, YSE is defined in the abstract but other abbreviations are not. It would improve clarity to define the abbreviations in lines 39-41 of the abstract i.e. ‘alleviated chronic ammonia-induced adverse impacts on fish growth by increasing final body weight (FBW), specific growth rate (SGR), feed intake and protein efficiency ratio (PER) while reducing feed conversion ratio (FCR) via improving food intake, elevating hepatic insulin-like growth factor (ILGF-1) and suppressing myostatin gene (MSTN) expression levels with the restoration of lipid reserves and switching on lipogenic’ Insulin-like growth factor is IGF in the list of abbreviations (line 62) but ILGF-1 in the abstract (line 40) Lines 142-144 Provide details of how fish were exposed to ammonia e.g. how was it added to the water and what range was it maintained within? Lines 174-175 ‘The blood samples were collected randomly from the caudal vein and divided into two parts.’ It is not clear what is meant by collected randomly. If it means the order in which fish had blood samples collected, I suggest writing this ‘The blood samples were collected from fish in random order from the caudal vein and divided into two parts.’ Lines 174-175 Specify the volume of blood collected from each fish Check Table 3. It may be altered due to formatting but the value for TAN in the Yucca group is unclear Some improvements are required in the use of English. For example, use of the term ‘in consistent’ is confusing throughout the discussion (e.g. line 458, 514, 571, 579). The authors appear to mean ‘is consistent with’ rather than inconsistent. Sentences should not commence with terms like in consistent, consistent with, side by side with, or in line with, because this is more point form notation. Sentences should commence with terms like ‘Our results are consistent with…’. Numbers from the reference list should not be primarily referred to in the text e.g. Line 549 - These results are in agreement with [84,85] who reported that ammonia exposure caused alteration of lipid metabolism in chickens and [86,87] who demonstrated this in aquatic animals. Numbers from the reference list should be used like line 554 i.e. ‘Furthermore, Madison et al. [88] stated that cortisol enhanced carbohydrates and’ Typographical errors Line 81 - human being should be human beings Line 84-85 - ‘In addition to, sewage effluents,’ there should be no comma after ‘In addition to’ Line 89 – ‘diffuse across gill membranes duo’ duo should be due Line 96 ‘cellular functions impairment’ should be impairment of cellular functions Lines 106-107 ‘an immense source of several bioactive constituents’ Remove the word several Line 110 ‘Yucca schidigera is a medicinal herb represents an important future ecofriendly’ Should read ‘Yucca schidigera is a medicinal herb which represents an important future ecofriendly’ Lines 111-112 ‘Yucca schidigera contains bioactive components include’ Include should be including Line 119 ‘all over the world that could be used as a suitable studying model concerning nutrition’ should read ‘all over the world that could be used as a suitable model for studying nutrition’ Line 177-178 ‘The second other half of blood sample was collected in an plain Eppendorf tube without anticoagulant’ should read ‘The other half was collected in a plain Eppendorf tube without anticoagulant’ Line 393 ‘down regulate in the hepatic IL1B, TNFα and HSP70 gene expression levels compared to’ should read ‘down regulation of hepatic IL1B, TNFα and HSP70 gene expression levels compared to’ Line 402 – spelling mistake ‘regardingt’ Line 416 ‘Here we detected that,’ There should not be a comma after the word that Line 428 ‘Moreover, In the present study’ The word ‘in’ should not be capitalised Line 429 ‘chronic ammonia exposure may resulted’ should read ‘chronic ammonia exposure may have resulted’ Line 431 ‘diminishing’ should be diminution Line 443 ‘In the same time’ should read ‘At the same time’ Line 447 ‘For explaining the growth performance and growth-related and’ Needs to be reworded Line 454 The word ‘more’ should not be capitalised Line 484 ‘alteration in inflammatory genes expression’ should read ‘alteration in inflammatory gene expression’ Line 493 The word ‘we’ should not be capitalised Line 494 ‘organs’ should be ‘organ’ Line 515 ‘demonistrated’ should be demonstrated Line 517 ‘this enhanced the cells evolved in’ should read ‘this enhanced the cells involved in’ Line 527 ‘this lower and inhibit the formation’ should read ‘this lowered and inhibited the formation’ Line 541 ‘significant’ should be ‘significantly’ Lines 545 and 546 ‘enzymes’ should be ‘enzyme’ Line 558 ‘may exerted’ should read ‘may exert’ Line 590 ‘elevated IGF-1 with suppressing the hepatic and brain MTSN expression’ should read ‘elevated IGF-1 suppressing hepatic and brain MTSN expression’

Author Response

Thank you for your valuable comments

  • Comments Whilst a list of abbreviations is provided after the abstract

Full names have been mentioned in abstract

  • Insulin-like growth factor is IGF in the list of abbreviations (line 62) but ILGF-1 in the abstract (line 40)

    Has been corrected to IGF-1  

  • Lines 142-144 Provide details of how fish were exposed to ammonia e.g. how was it added to the water and what range was it maintained within?

YSE was supplied every two days to ensure constant supply.

  • Lines 174-175 Specify the volume of blood collected from each fish

The volume of blood was variable as sometimes fish gives small amount and then we pool samples.

  • Check Table 3. It may be altered due to formatting but the value for TAN in the Yucca group is unclear

The values are clear but probably due to formatting it is not clear, in the tables file it is more clear.  

  • Some improvements are required in the use of English.

We have performed extensive English editing in MDPI English editing services and the certificate has been sent to the editor

All typing error have been corrected  

Reviewer 5 Report

The paper entitled “Exploring the Multimodal Role of Yucca schidigera Extract in  Protection against Chronic Ammonia Exposure Targeting:  Growth, Metabolic, Stress and Inflammatory Responses in Nile tilapia (Oreochromis niloticus L.): as Determined by Gene Expression” aims to demonstrate the beneficial role of YS on stress response and tolerance induced by ammonia. the topic is interesting for tilapia farming which is intensive and over exploited. Introduction provide a complete state of art of the topic, description of  M&M is complete and exhaustive, statistical analysis appropriate and the results are clearly presented and supported the hypothesis of the authors. the only aspect to improve is the discussion section which is certainly too long and appears as a list of works in accordance with what is highlighted in this work. this part should be rewritten trying to better argue and certainly shortening the text. In the same view the title should be shortened.

Author Response

Thank you for your valuable comments

The title has been shortened

Regarding the discussion section: The experiment contains a lot of parameters which must be all explained and discussed

Round 2

Reviewer 2 Report

The authors have ignored some of my commets though they are not major. Table 3 still lacks the superscripts.

Reviewer 3 Report

I would like to thank the authors for preparing revise version. However, the references and author contributions are still not according to guideline of Animals journal.